# Checklist of African Soapy Saponin—Rich Plants for Possible Use in Communities’ Response to Global Pandemics

**DOI:** 10.3390/plants10050842

**Published:** 2021-04-22

**Authors:** Yvonne Kunatsa, David R. Katerere

**Affiliations:** Department of Pharmaceutical Sciences, Tshwane University of Technology, Arcadia 0083, South Africa; yshonhe@gmail.com

**Keywords:** hand hygiene, antimicrobial activity, soapy plants, saponins, phytochemicals, toilet soap

## Abstract

Plants that exhibit foaming properties when agitated in aqueous solutions are commonly referred to as soapy plants, and they are used in different communities for washing, bathing, and hair shampooing. The frothing ability of these plants is attributed to saponins which are also well-documented to possess antimicrobial attributes. In the light of COVID-19, soap and hand hygiene have taken center stage. The pandemic has also revealed the low access to running water and commercial soaps in many marginalized and poor communities to the detriment of global health. Thus, soapy plants, either in their natural form or through incorporation in commercial products, may be a relevant additional weapon to assist communities to improve hand hygiene and contribute to curbing COVID-19 and other communicable infections. This review paper was compiled from a review of literature that was published between 1980 and 2020. We found 68 plant species, including those which are already used as traditional soaps. Our findings support the potential use of extracts from soapy plants because of their putative viricidal, bactericidal, and fungicidal activities for use in crude home-based formulations and possibly for developing natural commercial soap products.

## 1. Introduction

Most medicinal and health-promoting properties of plants are due to secondary metabolites which exist in forms and concentrations which vary with species of plants and plant parts. Geographical, environmental, and climatic conditions also affect the types and amounts of phytochemicals [1]. Some of the most common classes of phytochemicals are terpenoids, saponins, tannins, flavonoids, and alkaloids [2,3]. 

Modern toilet soaps and detergents trace their origin to the ancient use of plants, commonly referred to as soapy plants, which possess foaming ability when they are agitated in water. The soapy properties of these plants are attributed to the presence of saponins and their scientific nomenclature arose as a result of this; for example, the genus Saponaria is made up of saponin-rich plants commonly called soapworts [4]. However, there are other genera such as Sapindaceae [5,6], Aceraceae [7,8], and Hippocastanaceae [9,10], whose saponin concentrations are also high. Saponins are identified by their ability to rupture erythrocytes or form colloidal solutions that can produce a stable lather when they are shaken in the presence of water [11]. 

COVID-19, a disease caused by the novel coronavirus (SARS CoV-2) and which may be transmitted due to poor hand hygiene, was declared a pandemic by the World Health Organization (WHO) on 11 March 2020 [12]. By the end of 2020, over 81 million people had tested positive and over 1,7 million had died due to the pandemic [13]. The standard recommendations for reducing the risk of transmission are social distancing, wearing masks in public spaces, using hand sanitizers, and regular handwashing with soap and water.

The COVID-19 pandemic has highlighted the need to maintain personal hand hygiene, not only for this disease, but also for other infectious diseases. The WHO has reported that access to soap and water in 2019 was limited to only 57% of the world’s school children, with 818 million, that is 43%, having no access to soap and water [14]. In Kenya for instance, 78% of the population lacks piped water and the price of soap is US $0.20–$0.44, which is out of reach of a large section of the population [15]. Therefore, limited access to water and soap are regarded as major barriers to good hygiene, something which has now emerged to be a cornerstone to non-pharmaceutical interventions in combatting COVID-19 and future pandemics.

Various studies have been published that reported the use of plants for oral hygiene purposes [16,17]. Moreover, there is a surge of sites on the internet that focus on formulation of plant-based shampoos, and this is probably due to the increase in the demand for commercial shampoo [18]. However, there is very little literature on the use of plants for hand hygiene. This highlights the fact that this is a neglected area of plant utilization. It may be that there is an assumption that hand and toilet soap are accessible to everyone in the world. This is not the case because the UN reported that about three billion people (which is 40% of the world’s population) survive without basic handwashing facilities with soap and water available in their homes [19]. Thus, a focus on the application of ethnobotany to hand and toilet hygiene is a unique approach on the role plants can play in enhancing community health. This paper tries to integrate what is known about saponins, their utility as soaps and how they can be applied in the community, either directly by communities which have no access to commercial soap, or by companies or non-governmental organizations formulating soaps using local resources.

A possible strategy to increase soap availability and affordability would be the use of the floristic heritage in many poor and rural communities. Soapy plants present natural sources of soaps with putative antimicrobial and disinfectant properties [20] which can be used in their natural forms as extracts or formulated into finished low-cost products. In order to do this, we saw the need for an in-depth review of literature and synthesis of knowledge on saponin-rich plants generally and Southern African flora in particular. Thus, the scope of the review was to report on the occurrence and distribution, pharmacology and toxicity, mechanism of action, and overall availability of saponin-rich plants in Southern Africa.

## 2. Results

We found 103 articles which matched our criteria and were published in the past 40 years, of which 24 were review papers. Fifty-three (53) publications were removed because the 40 mg/g saponin threshold was not met. Therefore, the results for saponin content presented in this review were compiled using 26 research articles.

From the remaining studies that were reviewed in detail, 51 species were found which are distributed in 32 families. Of these species, only 15 species from 10 families have actually been reported as being used for soaps or shampoos in various communities.

### 2.1. Saponin-Rich Plants

As shown in Figure 1, saponins are high molecular weight organic molecules, which form glycone (saccharide) and aglycone (non-saccharide) moieties upon hydrolysis [21,22]. The glycone part of saponins consists of one or two sugar moieties. The saccharide moiety is usually in the form of either pentoses or hexoses and is water soluble. It has been noted that arabinose, xylose, glucose, ribose, glucuronic acid, and rhamnose are the most commonly occurring saccharide moieties in saponins [22,23]. The part which is non-soluble in water, also referred to as the sapogenin or genin, is characterized by 27 to 30 carbon atoms [18] and is either a triterpene or a steroid [22]. Depending on the type of aglycone that they have, saponins are classified into triterpenoid or steroid saponins, the former of which is more common in the plant kingdom [2].

Saponins are present in different plants, in different quantities ranging from low to high. Table 1 gives information on plants that were found to have various concentrations of saponins, through different quantification techniques, by different researchers around the world. It is also worth noting that a 40 mg/ g concentration of saponins in plants was the benchmark for inclusion in this review.

Apart from foaming properties, saponins are typically identified by their ability to exhibit hemolytic activity, which also renders them the ability to disrupt microbial cells. Thus, plants that exhibit high hemolytic activity are more likely to have more saponins. *Solanum macrocarpon* L. and *Balanites aegyptica* L. Delile have been reported to exhibit high hemolytic activity. The leaves of *S. macrocarpon* L. were tested on AA, AS, and SS types of red blood cells, and the SS blood cells were more susceptible to hemolysis than the AA and AS cells [50]. This shows that the level of hemolysis that is exhibited by plants also depends on the type of blood cells. Surprisingly, while the hemolytic activity of *S. macrocarpon* L. on AA blood cells was low, that of *B. aegyptica* L. Delile was high [51]. The hemolytic concentration of a plant extract also shows its hemolytic strength. Plants that exhibit hemolytic activity at lower concentration exhibit high hemolytic activity.

The type of extract also impacts on the hemolytic activity of the samples, as much as the type of plant does. There could be an interaction between the extraction solvent and the type of plant. In one study, *Callistemon viminalis* (Sol. Ex Gaertn) G. Don’ and *Kalanchoe pinnata* (Lam) Pers. were extracted with a range of solvents before being tested for their hemolytic activity. It was observed that the order of percentage hemolytic activity was not according to the types of solvents. The percentage order of hemolytic activity of *C. viminalis* (Sol. Ex Gaertn) G. Don’and was as follows: Chloroform > Ethyl-acetate > 90% methanol > 95 % methanol > Absolute methanol > Petroleum ether > n-butanol, while that of *K. pinnata* (Lam) Pers. was as follows: 95% methanol > n-butanol > Petroleum ether > Absolute methanol > Chloroform > Ethyl-acetate > 90% methanol [52].

As highlighted in Table 1, saponin-containing plants fall under a variety of taxa, though Fabaceae and Lamiaceae families were the most common in this review. We also noted that the quantities of saponins in plants still vary, even within families. There is evidence that environmental conditions, tissue type, age, physiological state, and genetic profiles of plants also impact on the concentration of phytochemicals present within a plant [22]. For plants such as legumes, biotic factors such as the presence of symbiotic mycorrhizal fungi enhance production of higher quantities of better-quality phytochemicals [53]. These principles also apply to saponin concentrations, resulting in varying concentrations in the same plant, from different geographical regions.

Saponin content in different parts of a single plant may also vary [46]. For example, the results presented in Table 1 show the differences in saponin content in aqueous extracts of various parts of *Eucalyptus camaldulensis* Dehnh as follows: roots (341 mg/g), bark (199 mg/g), fruits (171 mg/g), and leaves (97 mg/g). This shows that the saponin content of one plant part cannot be regarded as the value for all the parts of the same plant. The maturity of the plant parts is also an important factor that affects the concentration of saponins. As shown in Table 1, the saponin content values for the methanolic extracts of ripe and unripe peels of *Mangifera indica* L. were 214.15 mg/g and 159.50 mg/g, respectively. This shows that as the M. indica L. fruit ripens, the saponin concentrations also increase.

The results that are shown in Table 1 suggest that researchers use different solvents for extracting saponins, and this is more likely to affect the amount of quantifiable saponins that are extracted. The results for *E. camaldulensis* Dehnh also confirm variations in the saponin content when different solvents are used to extract the same part of the plant. The roots of *E. camaldulensis* Dehnh yielded 341 mg/g when they were extracted with water, and 261 mg/g with ethanol. Similarly, the aqueous extracts of the leaves had 97 mg/g saponin content, while the ethanolic extract had 125 mg/g. There is no particular formula in the saponin concentration changes, with respect to changes in solvents. In the two examples above, the aqueous extracts had higher quantities compared to the ethanolic extracts for the roots, and the opposite is true for the leaves. It is more probable that the variation could also be a factor of the hardness of the plant parts, considering that the fruits of the same plant showed the same downward direction of changes as the roots, as shown in Table 1. The leaves, which are relatively softer compared to the other studied parts, showed a rising change from aqueous to ethanol solvents.

Due to their unique structure and the different positional arrangements of sugar moieties, it is difficult to assign specific solvents and methods for extraction of saponins [54]. Methanol and ethanol, or the aqueous solutions of both, as well as 100% aqueous solutions, are commonly used as solvents for saponin extraction [55,56,57]. However, it is also, in most cases unclear whether the saponin values shown in Table 1 are triterpenoid or steroid saponins as they are usually simply termed ‘saponins’ in the literature. It is also important to note that the methods for extraction of saponins also vary, as researchers consider factors such as available equipment, speed and efficiency of extraction, as well as the resultant extraction yield. Methods such as maceration [58], ultrasonic-assisted extraction [59,60], Soxhlet extraction [60], low pressure refluxing [61], and microwave-assisted extraction [60,62] are being employed for extraction of saponins.

### 2.2. Occurrence and Distribution

The occurrence of soapy plants or saponin-containing plants has been reported in various parts of Africa, including the southern region of the continent. In Southern Africa, countries such as South Africa, Zimbabwe, Zambia, Mozambique, Lesotho, Namibia, and Swaziland are reported to contain a floristic wealth of saponin-containing plants. Some of the plants are endemic to these Southern African countries, whilst others migrated and naturalized from other African countries and abroad. Some of the reports available on the occurrence and distribution of soapy plants in Southern Africa are as follows: *Dicerocaryum zanguebarium* (Lourr.) Merill in Zimbabwe [63]; *Pouzolzia mixta* Sohms in Botswana, Malawi, South Africa, Swaziland, and Zimbabwe [64]; *Sesamum angolense* Welw in Zambia [65]; *Deinbollia xanthocarpa* (E. Mey.) Radlk in Malawi, Mozambique, South Africa, Zambia, and Zimbabwe [66] *Deinbollia oblongifolia* (E. Mey. ex Arn) Radlk in Mozambique and South Africa [67] *Albizia versicolor* Welw. ex Oliv in South Africa [68]; *Helinus integrifolia* (Lam) Kuntze in South Africa [69] and *Senna obtusifolia* L. in South Africa [70].

### 2.3. Traditional Uses

Some plants have been documented to be used traditionally as natural soaps and shampoos as shown in Table 2. Leaves, twigs, roots, stem barks, fruits, seeds, and flowers of these plants are rubbed or agitated in water, forming a stable lather, which is then used for washing, bathing, and hair shampooing. The nomenclature of these plants is usually based on their soapy qualities, such that even the common names usually include “soap” e.g., soap aloe, soapwort, soap tree, soap berry, soap bark, and soap dogwood. The families of saponin-rich species used in Southern Africa are Aloaceae, Fabaceae, Malvaceae, Pedaliaceae, Rhamnaceae, and Tiliaceae. The review shows that only a small sub-set of saponin-rich plant species are traditionally utilized. Table 2 shows saponin-rich plants that are being traditionally utilized, as compared to a larger number of saponin-rich plants, as shown by Table 1, that are not being used for their soapy characteristics. This may be because of lack of knowledge about, and access to, the plants.

### 2.4. Topical Use

Saponins have been reported to be active ingredients in some patented dermatological products which are topically applied for haircare. For example, Meybeck et al. patented a skin and haircare product which contained at least one saponin of the ginsenoside type [84]. Dermal toxicity tests that were done for saponins from *Sapindus mukorossi* Gaertn showed that they are not toxic to the skin [85]. Topical administration of saponins in guinea pigs was reported to cure skin lesions, which had been induced by infecting the guinea pigs with *Trichophyton mentagrophytes*, a fungus. A report from another study suggested that a saponin fraction from Korean red ginseng had great potential to treat atopic dermatitis. This was shown by the ability of the saponins to reduce clinical skin severity scores and ear thickness in mice which had induced atopic dermatitis-like lesions [86]. Therefore, based on the literature, saponins can be topically administered even on sensitive skin, and for example, Medicago saponins have been patented to enhance the renewal of the epidermis of the skin [87].

### 2.5. Studies to Support Antimicrobial Use

If soapy plants are to be used by communities as toilet soaps either in crude or processed/ commercial form, we were interested in finding out whether there is scientific data to support any antimicrobial claims which can be made.

Experimental work that reported antimicrobial activity of crude plant extracts was not considered in this review because crude extracts contain a greater variety of phytochemical compounds, other than saponins, compared to saponin-enriched extracts. Other phytochemicals such as alkaloids, or interactions between phytochemicals, may be responsible for some observed and reported antimicrobial activity in plants. Some studies have proven different levels of antiviral activity by alkaloids against a variety of viruses, both in vitro and in vivo [88,89,90,91,92]. Therefore, only the antimicrobial activity that is attributed to purified saponins and saponin-enriched extracts was reported in this review, as evidence of antimicrobial activity of saponins, particularly triterpenoid saponins which exist in abundance in soapy plants.

Numerous studies show evidence that saponins possess potent antiviral, [93,94,95,96,97,98], antibacterial, and antifungal activities [99,100,101,102].

The susceptibility to saponins by microorganisms may be due to the types of saponins as well as the types and structures of the microorganisms concerned. For example, enveloped viruses such as the influenza and corona viruses are more difficult to destroy because of the protective lipid bi-layered envelope that surrounds a shell of membrane associated proteins that are encoded by the virus [103]. Non-enveloped rotaviruses are more susceptible to xenobiotics because of the absence of the envelope. In addition, RNA viruses tend to mutate more than DNA viruses [104] and are therefore more difficult to treat. On the other hand, Gram-negative bacteria like *Escherichia coli* tend to be more resistant to antibacterial agents, compared to Gam-positive bacteria, because of their cell wall. Gram-negative bacteria have an external membrane that is absent in Gram-positive bacteria. This outer membrane provides the Gram-negative bacteria with extra protection against antimicrobial agents [105] as it thicker. Unlike the viruses and bacteria, fungi are eukaryotic, a factor which makes it difficult to have them treated because antifungal agents may also be toxic to mammalian cells.

From a use perspective, non-selective compounds, such as saponins, whose mode of action is cellular hemolysis, provide a wide spectrum of disinfection which is useful for hand and surfaces, as it is based on contact. This is suggested as the same method that saponins employ in their activity as deterrents and toxins against herbivores in the plant [106].

Saponins have the same chemical structure as soaps and detergents, equally consisting of hydrophilic heads and hydrophobic tails. The hydrophobic tails interact with the lipids in the envelopes of bacteria and viruses, causing them to disrupt and release their contents. The proteins from the disrupted envelopes are then engulfed in saponin molecules in the same way that soaps form micelles around dirt. Degradation of the cell wall of bacteria and subsequent destruction of the cell membranes of S. aureus, S. epidermidis, and *Bacillus cereus* by *Chenopodia quinoia* Willd saponins has been reported [100]. The interaction of saponins with cell membranes also causes leakage of cell contents in fungi, leading to their degradation [107].

Saponins interact with membranes in various ways, a characteristic which enables them to disturb the normal interactions between microorganisms and the cells of the human body. The saponins interact with the cholesterol in cell membranes. Cholesterol is usually part of the lipid rafts that play an important role in the entrance of microorganisms into host cells. Therefore, its absence as enhanced by its interaction with saponins may hinder the success of attack of human cells by microorganisms [108].

In another study, a triterpenoid saponin which was extracted and isolated from *Anagallis arvensis* L. was reported to show in vitro antiviral activity against the herpes simplex virus type 1 and poliovirus type 2. The saponin successfully inhibited the cytopathic effect of the virus, in addition to reducing viral replication. The mode of antiviral action of saponins was also proposed to be the inhibition of the attachment between the host cells and viruses [109].

A study by Simões et al. suggested that some saponins act by inhibiting the synthesis of capsid proteins of the target virus. In their research, they analyzed the activity of a triterpenoid saponin, which had been isolated from a Chinese plant (s17), against Herpes simplex virus 1. It was reported from the study that the saponins inhibited the synthesis of the capsid proteins of the virus [110].

### 2.6. Activity against Viruses

#### 2.6.1. Non-Enveloped Viruses

An in vivo experimental study on saponin extracts from *Quillaja saponaria* Mollina demonstrated antiviral activity of the extracts against rotavirus in mice. The antiviral activity was noted by monitoring changes in the diarrheal symptoms that were induced by the rotavirus. A 68% decrease in diarrheal symptoms was reported when a 0.015 mg per mouse dose of the saponin extract was administered to mice which had been pre-exposed to 500 plaque forming units of the rotavirus daily, for a five-day period [96].

#### 2.6.2. Enveloped Viruses

In an in vitro study that was done by Lee et al. (2012), a conclusion was reached that a concentration of 10 µg/mL of saponins inhibited the replication of the hepatitis C virus, in cultured, virus-infected cells. The study suggests that saponins can regulate genes in cells. It was also demonstrated in the same study that saponins significantly increased the protein level of suppressor cytokine signaling 2 (SOCS2), resulting in the inhibition of the replication of hepatitis C virus [95].

Purified saponins that were extracted from the leaves of *Quillaja brasiliensis* (A. St.-Hill. & Tul) Mart enhanced an experimental rabies vaccine’s effects by inducing specific immune response. The saponin-based adjuvant vaccine was tested in mice, against bovine Herpes virus type 1 and 5, bovine diarrhea virus, and poliovirus. *Q. brasiliensis* (A. St.-Hill. & Tul) saponins increased the levels of IgG significantly. It was concluded from the study that the saponin-incorporated vaccine protected the mice models from lethal exposure to rabies [93].

Tea seed saponins which were extracted from the seeds of *Camelia sinensis* L. were experimentally investigated for their activity against human type A and B influenza viruses. An effective inactivation of the viruses was reported at doses of 60 µg/mL for H3N2 virus, 80 µg/mL for B/Lee/40, and 100 µg/mL for the H1N1 virus. A dose-dependent inhibition of the influenza type A virus was also reported from the same study, within the concentration range of 1–30 µg/mL [93].

An investigation of the antiviral properties of triterpenoid saponins which were extracted from *Potentialla anserine* L., was carried out against Hepatitis B virus. The two-fold study was done in vitro in a 2.2.15 cell line and in vivo in Peking ducklings. The results which were reported from both experiments showed that triterpenoid saponins from P. anserine inactivated the Hepatis B virus by inhibiting the replication of its DNA. A decrease in the expression levels of HBsAg, HBeAg, and the DNA of HBV was reported from the in vitro experiments [96].

### 2.7. Activity against Bacteria

#### 2.7.1. Gram-Positive Bacteria

Saponin extracts that were prepared from *Sorghum bicolor* L Moench were tested for their antimicrobial activity against *Escherichia coli*, *Candida albicans*, and *Staphylococcus aureus*. The results showed inhibition of *S. aureus* [99].

In another experiment, the Gram-positive cocci, *Staphylococcus epidermidis* and *S. aureus* were susceptible to inhibition by saponins from *Chenopodium quinoia* Willd [100]. The Minimum Inhibitory Concentration (MIC) was 0.0625 mg/mL, while the Minimum Bactericidal Concentration (MBC) was 0.125 mg/mL, for both microorganisms. It was reported from the same study that the relationship between the *C. quinoia* Willd extracts and the inhibition of the microorganism was dose dependent.

#### 2.7.2. Gram-Negative Bacteria

Saponins that were extracted from *Solanum trilobatum* L. were tested for their antibacterial activity against *Escherichia coli, Pseudomonas aeruginosa,* and *Klebsiella pneumoniae*. The results from the study showed that the saponins inhibited the growth of all three Gram-negative bacteria, with *P. aeruginosa* being the most susceptible [111].

The antibacterial efficacy of saponins that were extracted from the leaves of *Tephrosia vogelii* Hook. f. was observed against *N. gonorrhea* and *E. coli.* It was observed in the same study that aqueous extracts were superior to methanolic extracts [112].

### 2.8. Activity against Fungi

Khanna and Khannabiran reported that *Aspergillus niger* and *Aspergillus flavus* were inhibited by the saponin extract from the roots of *Hemidesmus indicus* (L.) R. Br. They also observed that the saponin fraction exhibited more activity against these fungal strains than bacterial strains that they included in their study. These bacterial strains were *S. aureus*, *K. pneumoniae*, *S. typhi*, *P. mirablis*, *E. coli*, and *P. aeruginosa*. It is also worth noting that the zone of inhibition that was observed for a concentration of 10 mg/mL of the pure saponin extract from the roots of *H. indicus* (L.) R. Br. was the same as that of a 25 mg/mL concentration of Chloramphenicol [113].

Three saponins that were extracted from *Allium minutiflorum* Regel were reported to have significant antifungal activity against ten fungal species, which included *Alternaria alternata*, *Alternaria porri*, *Fusarium oxysporium*, *Trichoderma harziunum*, and *Pythium ultimum*. The results from the study suggested that *Trichoderma spp* were highly sensitive to *A. minutiflorum* Regel saponins. These results tallied with other studies that reported susceptibility of *Trichoderma* species to saponins from *Panus quinquefolius* L. and *Medicago sativa* L. [114,115].

Saponins are active against various types of fungi, including commercially important yeasts. Another piece of research revealed the antifungal properties against *Saccaromyces cerevisiae* by saponins from *M. sativa* L. and *Medicago arborea* L. [102]. Much more evidence on the antifungal activity of saponins has been reported from various studies [116,117,118,119].

### 2.9. Product Development

In thinking about product development, several parameters should be considered, e.g., availability and easy access to the raw material, the plant part which can be sustainably used, the state of the material (fresh or dried) and the type of formulation (crude extract or processed). Dry extracts may be preferable, but it should be noted that saponins are sensitive to heat [120]. Therefore, optimum temperatures which do not affect their stability should be identified during product development and processing, so that their efficacy is not reduced. Drying also improves stability as high humidity or water content enhances microbial spoilage. It is recommended that saponin extracts should be stored under conditions of low temperature and relative humidity to maintain their physico-chemical integrity [121].

Based on the above, the ideal commercial formulations should be dry soap powder or bars rather than liquid soap. Extemporaneous use with crude extracts is also viable as this usually entails short storage times.

### 2.10. Conservation Status

A legitimate concern in the promotion of biological resources is their conservation status, as widespread use can further imperil those species which might be red listed by the International Union for Conservation of Nature (IUCN). Most of the soapy plants highlighted in this review are to be found growing in the wild e.g., *D. zanguebarium* (Lourr.) Merrill*, N. Africana* L., and *L. camara* L. while a few species such as *Aloe* species, *M. oleifera* Lam, *J. curcas* L., and *G. max* (L.) Merr are being cultivated for other purposes.

The climatic conditions under which soapy plants survive are wide-ranging from arid and/or semi-arid to tropical wet climates. *X. americana* L.*, M. oleifera* Lam, and *J. curcas* L. grow well in arid and semi-arid climates [122]. Other plants, like *P. oleracea* L.*,* can thrive in a wide range of climates, temperatures, and soil types [123]. As such, saponin-containing plants are available all around the globe.

Most of the soapy plants which have been mentioned in this review are classified under the Least concern (LC) category by the IUCN’s Red List of threatened species. These are the following (the emboldened species are also cultivated): *Albizia versicolor* Welw. ex Oliv, *Saponaria officinalis* L., *Senna obtusifolia* L.*, Noltea Africana* L., *Portulaca oleracea* L.*, Erythrina senegalensis* DC, *Afzellia bella* Harm, *Annona squamosa* L., *Ageratum conyzoides* L., *Gmelina arborea* Roxb*, Vaccinium macrocarpon* Aiton*, Schotia latifolia* Jacq*, Hibiscus articulates* Hochst, exA. Rich *Monodora myrstica* (Gaertn) Dunal*, Ximenia Africana* L., *Dichrostachys cinerea* Wight et Arn., *Pouzolzia mixta* Sohms*, Foeniculum vulgare* Mill, *Sacoglottis gabonensis* (Baill.) Urb.*, Sapindus Saponaria* var. drummondii (Hook. & Arn.) L.D. Benson*, Taraxacum officinale* F. H. Wigg., *Jatropha curcas* L., *Persea americana* Mill., *Deinbolia oblongifolia* (E. Mey. Ex Arn) Radlk, and *Quillaja saponaria* Mollina.

*Mangifera indica* L. and *Kampferia galanga* L. were categorized under the Data deficient (DD) category because, according to the IUCN, the information available on these plants is inadequate for direct or indirect determination of whether the species are at risk of extinction*. Eucalyptus camaldulensis* Dehnh was the only plant in this review that was classified as a Near Threatened (NT) species. This implies that the plant is at risk of being threatened in the near future, hence the need to recommend conservation measures to reduce the chances of its extinction. The rest of the plants mentioned in this review were not categorized in any of the Red List categories. The use of invasive plants like *Lantana camara* would be a good strategy to contain their further spread.

It is worth noting that some of the plants are not known by members of various communities in which they are available. The effect of this is that there may be limited use of the plants even when they are available. Communities may need to be informed about the potential uses of the plant species and also offered training in their sustainable use, e.g., using leaves and fruits and not bark or underground parts which are destructive.

This is an important study because it advocates the application of scientific knowledge (i.e., saponin-containing plants, their scientific and common names) to the procedures that members of communities can follow in preparing their soaps and disinfectants. It further provides information on the availability of the plants in various communities, as well as their conservation and cultivation status.

### 2.11. Relevance of Soapy Plants in Community Health

After the emergence and spread of COVID-19, it became clear that soap and water are important in combating the spread of this disease, together with other infectious diseases. However, there are people in various communities that have limited access to soap and water, if any [15]. The inaccessibility of soap stems from financial limitations in communities, especially rural ones. The information that is provided in this review provides such communities with alternatives that they can use in place of commercial soap. Since soapy plants have possible antimicrobial activities [93,112,113,115], the sap of such plants can be used as disinfectants for hands and surfaces, for communities that have limited access to water. Some of the plant-based preparations of soap require little amounts of water, and this also helps to overcome the barrier of limited water supplies. For those with access to water but not soap, they can use these plants as their sources of soap. At the end of the day, community health is improved as access to soap is increased.

Water, soaps, and antimicrobial disinfectants have become the cornerstone for reducing the spread of and infection by COVID-19 in particular and maintaining personal hygiene in general, which is vital for fighting other communicable topical infections such as scabies. The role of water in fighting pathogens is based on its ability in reducing microbial bioburden on hands and surfaces so that they are not transmitted into the body through the mouth, eyes, ears, or nose. Washing hands or the whole body and food preparation surfaces and utensils is effective when running water is available, which is often a challenge in rural and informal urban communities.

Soaps enhance the removal of microorganisms and dirt from body and other surfaces because they denature the microorganisms and emulsify the dirt thus performing a dual role of disinfection and cleaning. However, commercial soaps are not affordable or accessible to many marginalized and poor communities thereby impacting on personal hygiene and community health. Saponin-rich plants may help to solve these problems at three levels; first, saponins confer to plants soap properties akin to commercial toilet soaps and detergents. Thus, they can be used to bath, wash hands, utensils, and surfaces, or even shampoo hair, just like any other soap. Second, the antimicrobial properties that were reported for saponins and other co-occurring phytochemicals, make the plants possible alternatives to commercial hand and surface disinfectants. Third, using soapy plants may reduce the amount of water that is required for disinfection since aqueous extractions of saponins may not require large amounts of water. In fact, the sap of some plants can be used as is and could be even more effective due to undiluted saponin concentrations.

Communities can therefore make positive use of the flora that are native to them, to better their health by protecting themselves and others from the spread of infectious diseases. From this review soapy plants have also been found to be relatively safe, exhibiting low dermal and ecological toxicity.

## 3. Materials and Methods

Major databases including Google scholar, Science Direct, and Web of Science were used to search for relevant peer reviewed publications between 1980 and 2020. We used the following key words “soapy plants”, “saponins”, “antiviral activity”, “antibacterial activity”, “antifungal activity”, “Sub-Saharan Africa”, “antimicrobial activity”, and “phytochemicals”.

We narrowed down the list of species by using the 40 mg/g saponin concentration benchmark as minimum inclusion criteria. For a single plant with different values for saponin concentrations, as published in different research, the highest value was selected for this review. The plants that aligned with these requirements were compiled as one set of results, presented as Table 1. For another set of results, which is presented as Table 2 (a compilation of plants that have always been used by various communities as soaps and shampoos), any evidence that a plant, either in part or whole, frothed when vigorously agitated in the presence of water, endorsed its inclusion in this review. The froth test method was chosen because that is typically how communities determine the soapy properties of the plants.

## 4. Conclusions

We suggest that the checklist of plants in this review may be used as sources of saponins in their natural forms, for exploitation of their antimicrobial activity, or for development of products with enhanced antimicrobial activity. With formulation studies and basic quality control protocols, these plant species can be an indispensable tool for fighting infectious diseases as topical products for sanitizing hands and surfaces and thus increasing access of hygiene products to a wider global population. Protocols for extraction and use of commonly occurring soapy plants would be useful for their rational use by communities.

For non-crop plant species, it is important that conservation should be borne in mind when the plants are used. Such strategies should include training on the use of leaves and fruits rather than destructive harvesting.

## Figures and Tables

**Figure 1 plants-10-00842-f001:**
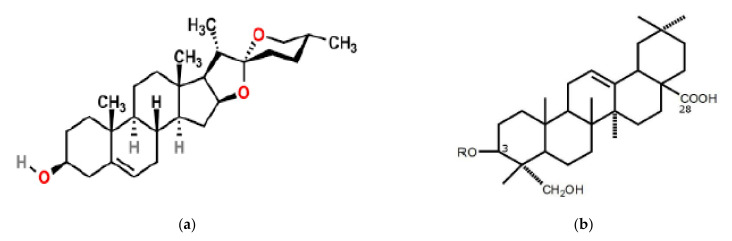
Possible structures of steroid saponins, with a steroid aglycone (**a**) (source [23]) and a triterpenoid aglycone (**b**) (source: [24]).

**Table 1 plants-10-00842-t001:** Plants which are rich in saponins as part of their phytochemical components.

Plant Name and Family	Common Name(s)	Geographical Location	Plant Part Used	Approximate Saponin Amounts (mg/g)	Type of Extract	References
Adoxaceae						
*Viburnum cotinifolium* D. Don	Smoke-tree leaved virbunum	Atlas Mountains (Northwest Africa)	Leaves	45.30	Aqueous ethanol	[25]
Aizoaceae						
*Carpobrotus edulis* (L.) N. E. Br.	Sour fig, ice plant	South Africa	Leaves, stems	45.00	Ethanol	[26,27]
Amaranthaceae						
*Amaranthus hybridus* L.	Pigweed	Southern Africa	Stem, leaves	184.00	Not stated	[28]
*Spinacia oleracea* L.	Spinach	Lesotho, Highveld of Southern Africa	Leaves	52.70	Methanol	[29]
Anacardiceaceae						
*Mangifera indica* L.	Mango	Southern Africa	Ripe peels	214.15	Methanol	[30]
			Unripe peels	159.50		
Annonaceae						
*Annona squamosa* L.	Sugar apple	Madagascar, Malawi, Mozambique	Fruit	63.88	Aqueous	[31]
*Monodora myristica* (Gaertn) Dunal	African nutmeg	Western and Eastern Africa	Seeds	120.40	Not stated	[32]
Apiaceae						
*Foeniculum vulgare* Mill	Fennel	South Africa, Zimbabwe, Ethiopia, Eastern Africa	Leaves	47.68	Aqueous	[31]
Apocynaceae						
*Pergularia tomentosa* L.		Horn of Africa	Leaves	44.40	Chloroform fraction	[33]
Asphodelaceae						
*Aloe ferox Mill*	Bitter aloe, red aloe, Cape aloe	South Africa, Lesotho, Southern Africa	Roots	41.20	Aqueous	[34]
Asteraceae						
*Ageratum conyzoides* L.	Billy-goat weed	Southern and Western Africa	Leaves	65.10	20% aqueous ethanol	[35]
*Taraxacum officinale* F.H. Wigg	Dandelion	Southern and Northern Africa	Leaves	50.60	Aqueous ethanol	[25]
Caesalpiniaceae						
*Piliostigma reticulatum* (GC) Hochst	Camel’s foot	Throughout Africa	Leaves	615.00	Methanolic	[36]
Combretaceae						
*Anogeissus leiocarpus* (DC.) Guill. and Perr.	African birch, Bambara	Eastern and Western Africa	Softwood	125.00	Not stated	[37]
Ericaceae						
*Vaccinium macrocarpon* Aiton	Large cranberry	Eastern and Southern Africa	Seeds	98.47	Acetone	[31]
Euporbiaceae						
*Jatropha curcas* L.	Physic nut	Throughout Africa	Leaves	800.00	Ethyl acetate fraction	[38]
Euphorbiaceae						
*Euphorbia hirta* L.	Spurge, asthma plant	Southern and Tropical Africa	Whole plant	400.90	Methanolic	[36]
Fabaceae						
*Afzelia bella* Harm	Afzelia	Central and Tropical Africa	Softwood	58.00	Not stated	[37]
*Dichrostachys cinerea* (L.) Wight & Arn	Sickle bush	Throughout Africa	Softwood	98.00	Not stated	[37]
*Erythrina senegalensis* DC	Coral tree, coral flower	Western Africa	Stem	344.40	Methanolic	[36]
*Glycine max* (L.) Merr	Soybean, soya bean	Sub-Saharan Africa	Seeds	184.00	Not stated	[39]
*Trigonella foenum-graecum* L.	Greek hay, Greek clover	Northern Africa	Seeds	506.90	Methanolic	[29]
*Schotia latifolia* Jacq	Boer-bean	Southern Africa	Stem bark	68.00	Aqueous	[40]
Humiriaceae						
*Sacoglottis gabonensis* (Baill.) Urb.	Bitter bark tree	Tropical Africa	Softwood	66.00	Not stated	[37]
Lamiaceae						
*Clerodendrum volubile* P Baeuv	Magic leaf	Western and Southern Africa	Leaves	136.70	Aqueous	[41]
*Coleus aromaticus* Benth	Indian borage	Southern and Eastern Africa	Leaves	62.30	Not stated	[42]
*Gmelina arborea* Roxb	White teak	Tropical Africa	Leaves	57.30	Methanolic	[29]
*Hyptis spicigera* Lam	Black sesame	Western, Central and Southern Africa	Leaves	62.30	Not stated	[43]
*Leucas linifolia* (Roth) Spreng	Dronpushpi	Africa	Leaves	48.20	Not stated	[42]
*Pogostemon patchouli* (Blanco) Benth	Patchouli	Throughout Africa	Leaves	142.30	Not stated	[42]
Malvaceae						
*Corchorus olitorius* L.	Jute mallow, Jew’s mallow, bush okra	Tropical Africa, Southern Africa	Leaves	43.00	Methanolic	[44]
*Hibiscus articulatus* Hochst, exA. Rich	Comfort root	Throughout Africa	Leaves	75.00	Methanolic	[44]
Menispermaceae						
*Cissampelos mucronata* A. Rich.	Hairy heartleaf	Tropical Africa and Southern Africa	Roots	446.70	Methanolic	[36]
Moringaceae						
*Moringa oleifera* Lam	Drumstick tree, horse radish tree	Eastern and Southern Africa	Softwoods	42.00	Not stated	[37]
Myrsinaceae						
*Myrsine africana* L.	African boxwood, Cape myrtle	Southern Africa	Fruits	175.00	20% aqueous ethanol	[45]
Myrtaceae						[46]
*Eucalyptus camaldulensis* Dehnh	Red gum, river red gum	Throughout Africa	Bark	199.00	Aqueous	
			Roots	341.00	Aqueous	
				261.00	Ethanolic	
			Leaves	97.00	Aqueous	
				125.00	Ethanolic	
			Fruits	171.00	Aqueous	
				82.00	Ethanolic	
Olacaceae						
*Ximenia americana* L.	Tallow wood, hog plum, sea lemon	Western and Southern Africa	Stem	508.60	Methanolic	[36]
Pedaliaceae						
*Dicerocaryum zanguebarium* (Lourr.) Merrill	Boot protectors, devil’s thorn	Southern Africa	Leaves	50.00	Methanolic	[44]
Portulacaceae						
*Portulaca oleracea* L.	Purslane	Tropical and Southern Africa	Aerial parts	320.00	Aqueous	[47]
Rubiaceae						
*Mitracarpus scaber* Zucc	Mutton grass	Angola, Northern	Leaves	43.20	Chloroform fraction	[33]
		Zambia, Malawi				
*Sarcocephalus latifolius* (Smith)	African peach	Tropical Africa	Leaves	481.80	Not stated	[38]
Sapindaceae						
*Sapindus emarginatus* Vahl	Notched leaf soap nut	Eastern tropics of Africa	Fruit	151.60	Aqueous	[48]
				180.40	Methanol	[48]
*Sapindus Saponaria* var. drummondii (Hook. & Arn.) L.D. Benson	Soap berry	Tropical and Southern Africa		212.50	Ethanol	[48]
				157.32	Aqueous	[31]
Scrophulariaceae						
*Butyrospermum paradoxum* (Gaertn, f.) Hepper	Shea butter	Sub-Saharan Africa	Leaves	838.90	Methanolic	[36]
*Striga hermonthica* (Del.) Benth.	Purple witchweed, giant witchweed	Sub-Saharan Africa	Aerial parts	307.90	Methanolic	[36]
Sterculiaceae						
*Waltheria indica* L.	Sleepy morning	Southern and Tropical Africa	Leaves	653.10	Methanolic	[36]
Verbanaceae						
*Clerodendrum colebrookianium* Walp	East Indian glory bower	South Africa	Leaves	88.00	Aqueous ethanol	[49]
*Lantana camara* L.	Tick berry, wild sage, red sage	Eastern and Southern Africa	Leaves	121.00	Not stated	[50]
Zingiberaceae						
*Kaempferia galanga* L.	Aromatic ginger, sand ginger	South Africa, Africa	Softwood	62.00	Not stated	[37]
*Zingiber cassumunar* Roxb	Cassumunar ginger	South Africa, Africa	Rhizome	69.03	Aqueous ethanol	[49]
Zygophyllaceae						
*Peganum harmala* L.	Wild rue	Southern Africa	Not stated	48.00	Aqueous ethanol	[25]

**Table 2 plants-10-00842-t002:** Plants that have been, and are still, used as soaps and shampoos by various communities.

Family and Scientific Name	English Common Name	Plant Part	Preparation and Use(s)	References
Aloaceae				[71]
*Aloe maculata* All	Soap Aloe	Leaves	The sap from the leaves is used as a soap for bathing and washing hair.	
*Aloe Saponaria* Mill	Soap Aloe	Leaves	The sap is used as soap for bathing.	
Caryophyllaceae				
*Saponaria officinalis* L.	Soapwort	Leaves	The leaves of the plant are added to pre-boiled water and left to simmer for about 5 min.	[72,73,74]
Fabaceae				
*Acacia concinna* Linn	Soap pod tree	Pods, bark	Roots that are boiled with water are used as soap. The dried and crushed bark forms a powder which is used as soap.	[75,76]
*Albizia versicolor* Welw. Ex Oliv	Large-leaved false thorn	Root, bark		[69]
Malvaceae				
*Sida rhombifolia* L (Bhuinli)	Mallows. fanpetals	Tender shoot bark	The tender shoot bark is rubbed on the skin or hair to produce lather during bathing and shampooing.	[77]
Pedaliaceae				
*Dicerocaryum eriocarpum* (Decne.) Abels.	Devil’s thorn, boot protectors	Flowers	The flowers are soaked in water to produce soapy water.	[78]
*Sesamum angolense* Welw		Leaves	An infusion of the leaves is used as soap for bathing and shampooing.	[66]
Quillajaceae				
*Quillaja saponaria* Mollina	Soap bark	Bark	The inner bark is reduced to powder and used as a soap.	[79]
Rhamnaceae				
*Noltea africana* L.	Soap dogwood, soap bush	Leaves, twigs	The leaves and twigs are rubbed in water to produce foam and the water is used for washing.	[80]
*Helinus integrifolius* (Lam) Kuntze	Soap creeper	Whole plant	The plant is infused in cold water, and a stick is used to agitate the water to produce lather.	[70]
Sapindaceae				
*Sapindus mukorossi* Gaertn	Soap nut, soap berry, wash nut	Fruit	The lather from the fruit is used as a soap for bathing and shampooing.	[79]
*Deinbollia oblongifolia* (E. Mey. Ex Arn) Radlk	Dune soap berry	Seeds	The seeds are lathered in water to produce soap.	[81]
Tiliaceae				
*Grewia ferruginea* Hochst ex. A. Rich	-	Leaves	The ash from the burnt leaves is used as soap.	[82]
Urticaceae				
*Pouzolzia mixta* Sohms	Soap nettle	Leaves	The fresh leaves are crushed and agitated in water to form a soap, which is used for bathing and washing.	[83]

## Data Availability

Not applicable.

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
