# Peer review of "Checklist of African Soapy Saponin—Rich Plants for Possible Use in Communities’ Response to Global Pandemics"

_plants, 2021, doi:10.3390/plants10050842_

Round 1

Reviewer 1 Report

This article concerned an overview of soap plants which could be used to improve the hygiene of african population during a pandemic periode. 

In scientifical point of view, this is an huge overview of what could be used with interesting propreties, but it is done with enrichement mixture of saponins extracted from plant samples. I think to have a good list, this article should be more focused on the antiviricidal properties for orl respiratory disaeses. All the properties should be evaluated from the crude extract like the leave, roots ...

A distributing duality is present because the answer to what kind of soap plant we could use for covid-19 pandemic is not present. All are focused on antimicrobial activites. In this way COVID-19 virus should not present in the keywords.

An ecology question could be a consequence of this review. If the plants are cultivated for this application what should be the consequence on the ecology? This is not an easy to consider.

There are some revisions to do to the perspectives of this article.

Author Response

Point 1:           In scientifical point of view, this is a huge overview of what could be used with interesting properties, but it is done with enrichment mixture of saponins extracted from plant samples. I think to have a good list, this article should be more focused on the antiviricidal properties for all respiratory diseases. All the properties should be evaluated from the crude extract like the leave, roots.

                        Response

                        First, our focus for this paper was on saponins only, because they are the source of the soapy properties that soapy plants have. Therefore, if we focus on crude extracts, we would not be sure if the antimicrobial activity that the soapy plants have would be from the saponins since there are other phytochemicals that have also been reported to have antimicrobial properties. This is the reason why we opted for saponin-enriched extracts so that we can tag the results to saponins. May you please see lines 185-194 in the manuscript.

                        Second, thank you for the idea of compiling plants that have antiviricidal properties against all respiratory diseases. We look forward to including the idea in our future work. However, for this paper, we looked at communicable diseases that are spread through physical contact and compromised hygiene, especially those whose spread can be countered by using soaps. In that regard, we are looking not only on diseases that are caused by viruses, but even those that are caused by bacteria and fungi.

Point 2:           A distributing duality is present because the answer to what kind of soap plant we could use for covid-19 pandemic is not present. All are focused on antimicrobial activities. In this way COVID-19 virus should not present in the keywords.

                        Response

                        The term “COVID-19” has been removed from the keywords. However, the focus was not on finding a plant soap that is specifically for the COVID-19 pandemic. The focus was on identifying various soapy plants that could be used just in the same way people use the commercially available soaps. This was based on the idea that at the moment, soaps are already being used as one of the major tools for fighting against the pandemic. In much the same way, soapy plants could be one of those soaps, contributing to the fact that members of various communities who have no access to commercial soap for one or the other reason, can use the soapy plants to wash their hands and disinfect surfaces.

Point 3:           An ecology question could be a consequence of this review. If the plants are cultivated for this application what should be the consequence on the ecology? This is not an easy to consider.

                        Response

                        Some of the plants that have been mentioned in this paper are already cultivated, and it is a matter of using by-products from them for soap making. There will be issues of growing plants for medicinal use, for food, and other varied uses, however, a balance has to be struck as the health benefits of these plants can go a long way in sustaining livelihoods. This information has been also highlighted in the conclusion on lines 426-428.

                        For some of the cultivated plants, parts that are normally thrown away after other uses of the plants are used. For example, mango peels are usually thrown away after eating the pulp of the fruit, and these have high saponin content which can be harnessed and used in the form of a soap.

                        Lastly, but not least, most of the plants that were mentioned in this paper are wild and self-existing. It is, however, recommended that people be taught to use the plants conservatively. For example, they can be taught not to cut the whole plant off so that it will keep existing (See line 429-431).

Reviewer 2 Report

This review was designed in a timely manner, and reference was well organized and comparative analysis was conducted. Readers will be able to gain knowledge about the types and applications of soapy plants based on the review contents. Nevertheless, it is thought that the editor's judgment is needed to determine whether the level of data collection or analysis method is appropriate for this “Plants” journal.

Author Response

Manuscript Number: Plants-1089813

Manuscript Title: Checklist of African soapy plants for possible use in communities’    response to global pandemics

Journal: Plants

REVIEWER 2

Point 1:           Nevertheless, it is thought that the editor's judgment is needed to determine whether the level of data collection or analysis method is appropriate for this “Plants” journal.

                        Response

                        We were invited to submit a paper on ethnobotany in community health and we believe that we have put together something which not only relevant and appropriate to the audience of this special edition, but also important to a wider audience.

Reviewer 3 Report

This is a mostly perfect work and in time paper to be published in Plants-MDPI. However, chemical structures should be provided again in more clear and readable structure. Secondly, they concentrate much on saponins and other biological activities and soap use, so please think again on the word "saponin" in the tittle. The search from plants and relevant information for soap use using Google and others are no need to write on the Abstract, please rewrite without those words.

Author Response

Point 1:           However, chemical structures should be provided again in more clear and readable structure.

                        Response

                        The chemical structures have been changed.

Point 2:           Secondly, they concentrate much on saponins and other biological activities and soap use, so please think again on the word "saponin" in the tittle.

Response

The term “saponins” has been included in the title, which now reads as follows: Checklist of African saponin - rich plants  for possible use in communities’ response to global pandemics.

Point 3:           The search from plants and relevant information for soap use using Google and others are no need to write on the Abstract, please rewrite without those words.

Response

The words were removed, and the statement now reads: This review paper was compiled from review of literature that was published between 1980 and 2020.

Round 2

Reviewer 2 Report

By reply of the author, they were invited to submit a paper on ethnobotany in community health and they believe that this special edition will be important to relevant researchers or general audience, I agree to accept this review.

Author Response

We have noted the Rev 2 response and it is in order.

Thank you